# Impact of Cancer Cachexia on Cardiac and Skeletal Muscle: Role of Exercise Training

**DOI:** 10.3390/cancers14020342

**Published:** 2022-01-11

**Authors:** Cláudia Bordignon, Bethânia S. dos Santos, Daniela D. Rosa

**Affiliations:** 1Oncology Center, Hospital Moinhos de Vento, Porto Alegre 90560-030, Brazil; cbordignon88@gmail.com; 2Graduate Program in Pathology, Federal University of Health Sciences of Porto Alegre (UFCSPA), Porto Alegre 90050-070, Brazil; 3Department of Clinical Research, Brazilian National Cancer Institute (INCA), Rio de Janeiro 20560-121, Brazil; bs_306@hotmail.com; 4Rede D’Or São Luiz, Rio de Janeiro 22271-110, Brazil; 5Brazilian Breast Cancer Study Group (GBECAM), Porto Alegre 90619-900, Brazil

**Keywords:** cancer cachexia, muscle wasting, cardiac muscle wasting, physical exercise

## Abstract

**Simple Summary:**

Cachexia is a syndrome that can be present in many patients diagnosed with cancer, especially in those with metastatic or very advanced tumors. The patient may present with weight loss, loss of muscle mass, and even cardiac dysfunction as a result of it. The aim of this review is to understand how cachexia manifests and whether physical exercise has any role in trying to prevent or reverse this syndrome in cancer patients.

**Abstract:**

Cachexia is a multifactorial syndrome that presents with, among other characteristics, progressive loss of muscle mass and anti-cardiac remodeling effect that may lead to heart failure. This condition affects about 80% of patients with advanced cancer and contributes to worsening patients’ tolerance to anticancer treatments and to their premature death. Its pathogenesis involves an imbalance in metabolic homeostasis, with increased catabolism and inflammatory cytokines levels, leading to proteolysis and lipolysis, with insufficient food intake. A multimodal approach is indicated for patients with cachexia, with the aim of reducing the speed of muscle wasting and improving their quality of life, which may include nutritional, physical, pharmacologic, and psychological support. This review aims to outline the mechanisms of muscle loss, as well as to evaluate the current clinical evidence of the use of physical exercise in patients with cachexia.

## 1. Introduction

Cachexia is a debilitating multifactorial syndrome, involving several metabolic pathways in different tissues and organs. It is characterized by systemic inflammation with progressive weight loss, depletion of adipose tissue, and loss of muscle mass that cannot be reversed by conventional nutritional support [1]. This condition can negatively affect patients during cancer treatment, reducing their tolerance and response to anticancer therapies, worsening their quality of life, and increasing mortality in patients with advanced cancer.

The prevalence of cachexia varies according to the primary site of the cancer. It is estimated to be higher than 80% for patients with pancreatic and gastric cancer, 61% for patients with colon, lung, prostate cancer, and non-Hodgkin lymphoma, and around 40% for patients with breast cancer, sarcoma, and leukemia. Overall, cachexia is indirectly responsible for 20% of all cancer-related deaths [2].

Besides the primary site of tumor, factors influencing cachexia origin and development are tumor histology, stage, and individual variations such as genetic predisposition, initial body mass index (BMI), and comorbidities. Cancer cachexia is defined as weight loss greater than 5% in the previous 6 months or corresponding to 2–5% for patients with either a BMI ≤ 20 kg/m^2^ or a reduced muscle mass (sarcopenia) [3].

Muscle wasting is the most important phenotypic feature of cancer cachexia. However, as cachexia is a multi-organ syndrome, in addition to the progressive loss of skeletal muscle mass, it affects other organs such as intestine, heart, kidneys, and liver [4]. In fact, the final cause of death in patients with cancer cachexia can be, apart from tumoral progression, heart arrhythmias, hypoventilation, thromboembolic events, and cardio-renal alterations.

Cardiac dysfunction is present in a substantial proportion of cancer cachexia-induced deaths [5]. Understanding the complex crosstalk between the heart and the skeletal muscle is crucial to establish the biological basis of cancer-induced cardiac cachexia and to propose potential targets for its therapy and prevention.

Physical exercise appears as an important strategy to counteract cancer cachexia-induced heart failure [6]. Indeed, aerobic training has anti-inflammatory effects, increases antioxidant defenses, prevents atrophy, and promotes oxidative metabolism.

In this review, we discuss the importance of better understanding the structural and metabolic changes within skeletal and heart muscles during cachexia in cancer patients. Besides, we evaluate the protective and therapeutic effects of physical exercise in cancer-induced cachexia.

## 2. Mechanisms/Pathophysiology

### 2.1. Altered Energy Balance

In normal conditions, a balance between the synthesis and the degradation of proteins maintains muscle homeostasis. An imbalance of these processes can lead to skeletal muscle wasting. In cancer-induced cachexia, the impairment of protein and energy balance results from a combination of reduced appetite with insufficient nutritional intake and metabolic dysfunction characterized by the catabolism of lean body mass and adipose tissue [7].

It is well known that tumors compete with other organs and tissues for energy fuels [8], leading to increased calorie wasting and contributing to metabolic unbalance. As muscles are a reservoir of amino acids that may be released for energy production during catabolic processes, some muscle loss induced by cancer-induced cachexia is expected.

In addition to alterations in protein metabolism, inflammatory cytokines are secreted by both immune and tumor cells. Distinct intracellular pathways, activated by inflammatory mediators, are associated with muscle protein turnover contributing to the muscle wasting process [9].

### 2.2. Tumor-Driven Inflammation

During cancer cachexia, there are multiple causes for the occurrence of inflammation, including the secretion by tumor cells and activated immune cells of different mediators, such as pro-inflammatory cytokines and inflammatory molecules, particularly, Tumor Necrosis Factor (TNF)-α, Interleukin (IL)-6, and IL-1 [10]. These cytokines, consequently, promote the activation of transcription factors associated with cachexia wasting in both adipose tissue and skeletal muscle.

TNF-α is considered as the major mediator of acute-phase proteins like C-reactive protein and fibrinogen. It is involved in muscle protein breakdown and large secretion of amino acids, mainly alanine and glutamine, from skeletal muscle [10]. In addition to mediate inflammatory response and proteolysis, TNF-α also activates nuclear factor-kappa B (NF-κB), with a rapid, dose-dependent response involving the phosphorylation and proteasomal degradation of the NF-κB-inhibitory protein IkBa [11]. The concentration of NF-κB is increased in the adipose tissue of patients with cancer cachexia, suggesting that the NF-κB pathway has a role in the promotion of white adipose tissue inflammation during cachexia. The increase of protein degradation via NF-κB affects the transcription of genes which regulate the ubiquitin–proteasome pathway (UPP) and promote proteolysis of myofibrillar proteins [12].

In addition, IL-1 induces anorectic and pyrogenic effects by increasing tryptophan plasma concentrations, leading to increased serotonin levels and causing early satiety and suppression of appetite. Furthermore, IL-1 is involved in two established pathways, the NF-κB pathway and the p38 and mitogen-activated protein kinase (MAPK) pathway [13]. The pro-inflammatory cytokine IL-6 activates the signal transducer and activator of transcription 3 (STAT3) pathway in skeletal muscle, resulting in atrophy.

## 3. Skeletal Muscle Wasting during Cachexia

In recent years, it has become clear that catabolic factors (members of the ubiquitin–proteasome system, myostatin, and apoptosis-inducing factors) are upregulated, while anabolic factors (such as insulin-like growth factor 1) are downregulated in cachexia muscle wasting [14]. The wasting of muscle mass results from a disturbed balance between anabolic and catabolic pathways. Three main catabolic pathways have been described in skeletal muscle to account for protein degradation, that is, ubiquitin-mediated proteasome degradation (UPR), autophagy, and signaling involving calcium-activated proteases such as calpain [15].

The autophagy–lysosomal (ALP) pathway and UPP degrade proteins that have been ubiquitylated by E1, E2, and E3 enzymes. In ALP, the ubiquitylated proteins are engulfed by autophagosomes and thereafter fuse to lysosomes to form autolysosomes, where proteins are enzymatically degraded [16]. UPP is upregulated through the expression of key E3 ligases such as atrogin 1 and muscle RING finger-containing protein 1 (MURF1). The expression of these E3 ligases mediate the degradation of structural muscle proteins including myofibrillar components [16,17]. In addition to the muscle E3 ubiquitin ligase genes, activated forkhead box O (FOXO) transcription factors have a vital role in transcribing genes involved in the autophagy system. Calcium-activated proteases have also been associated with the initiation of protein breakdown during cachexia.

In physiological conditions, skeletal muscle homeostasis requires autophagy to eliminate damaged proteins and organelles. However, the upregulation of autophagy genes leads to excessive activation of autophagy pathways that increase the breakdown of skeletal muscle. Mitochondria (and other organelles) are also degraded by autophagy, and their loss accounts for the decreased endurance capacity of atrophied muscles [15,18].

Inflammatory mediators, myostatin and proteolysis-inducing factor (PIF), activate intracellular signals. Cytokines and PIF, through transcription factors such as NF-κB and FOXO family members, control the expression of regulators of the autophagy pathway: MAFbx (also known as atrogin 1) and MuRF1 [2,19]. Furthermore, the p38 and JAK/MAPK cascades may also be activated by the same cytokines and PIF, in turn activating caspases and consequently, apoptosis. Myostatin, which acts through activin receptor type IIB (ACTRIIB)-mediated signaling, can also induce protein degradation by FOXO and trigger apoptosis via the MAPK cascade. The myostatin/activin pathway negatively regulates muscle size through the phosphorylation of SMAD2/3, primarily by inhibiting protein kinase b (Akt) [2] (Figure 1).

In parallel with the activation of catabolic pathways, there is an inhibition of anabolic processes, including reduced phosphatidylinositol 3-kinase (PI3K)–Akt signaling and mammalian target of rapamycin (mTOR)-dependent protein synthesis. This happens, at least in part, because of reduced insulin-like growth factor-1 (IGF-1) levels and the development of insulin resistance [20].

## 4. Cardiac Dysfunction Induced by Cancer Cachexia

In addition to skeletal muscle wasting, cancer cachexia is also associated with significant heart muscle dysfunction. Cardiac abnormalities are commonly found in cancer patients and represent the primary cause of death in at least one-third of cancer patients. Patients with cancer often present with clinical indicators of chronic heart failure, including fatigue, shortness of breath, and impaired exercise tolerance [1,5]. However, identifying the mechanisms of the development of cardiovascular complications in cancer patients has been challenging. The cardiac alterations may be due to underlying heart disease unrelated to cancer exacerbated by the treatment and cardiotoxic effects of the therapy or induced by the cancer itself.

Although the cachectic condition primarily affects the skeletal muscle, cachexia is considered a multi-organ disease that involves different tissues, including the heart. Cardiac cachexia is described as cardiac atrophy, remodeling, and dysfunction associated with cancer.

As previously described in the pathogenesis of skeletal muscle loss, cardiac muscular damage also involves metabolic alterations, primarily, increased in energy expenditure, proteolysis by the activation of the ubiquitin-mediated proteasome degradation UPR system, and autophagy [4,21,22]. Sequestration of nutrients by the tumor and metabolic changes in other organs are also involved in the cardiac cachexia process. For instance, the generation of proteins linked to the systemic inflammatory state by the liver has been proposed to contribute to energy wasting in cancer patients. Additionally, cancer cells can affect insulin and glucose metabolism, inducing insulin resistance by prompting the production of insulin-like growth factor (IGF)-binding protein 1 and negatively affecting plasma glucose levels. Tumor growth can directly impact on circadian rhythms, an alteration that is functionally linked to the onset of insulin resistance [21,22,23].

It is well known that cancer may activate many protein degradation pathways and inhibit protein synthesis pathways, leading to increased protein loss and muscular atrophy.

Finally, the signaling pathway that seems to be most involved in cardiac atrophy is protein degradation mediated by the expression of the transcription factor nuclear-κB (NF-κB). Pro-inflammatory cytokines, such as IL-6, were found in mice cachectic hearts and are associated with the activation of host intracellular pathways including NF-κB and caspases.

Although cancer-related cardiac atrophy needs to be more thoroughly elucidated in humans with cancer-associated cachexia, research in animal models has shown substantial cardiac atrophy in multiple cachexia-inducing tumors. In mice, cardiac atrophy has been associated with growth inhibition through the activation of ACTRIIB mediated by signaling molecules called myokines and cardiokines, which include members of the transforming growth factor (TGF) superfamily, like myostatin, activin A, and growth/differentiation factor 11 (GDF11) [5,21,22,23,24,25,26]. These findings reinforce the role of pro-inflammatory cytokines, such as IL-1, IL-6, and TNF, in mediating cardiac dysfunction.

## 5. Main Effects of Exercises in Muscle

### 5.1. Inflammation

Low-grade chronic inflammation is present in several pathologies, such as obesity, cardiovascular diseases, diabetes mellitus, and cancer [27,28]. In colorectal tumors, for example, systemic inflammation is associated with a worse prognosis [28]. Reducing inflammation associated with cancer-induced cachexia may be a target in controlling this manifestation.

The anti-inflammatory effect of physical training has been proved in recent studies. It is known that in the acute phase of physical exercise, IL-6 levels are increased by 100 times [28,29,30,31]. Although IL-6 is involved in pro-inflammatory events such as sepsis, during exercise it acts as an anti-inflammatory promoter. IL-6 induced by physical activity is considered a myokine, stimulates the production of IL-1ra and IL-10 (which are anti-inflammatory cytokines), and inhibits TNF-alpha, a pro-inflammatory cytokine [28,31].

Regular physical training can probably reduce inflammation present in many chronic diseases, according to currently available evidence [32]. In cancer-induced cachexia, physical exercise would also have the role of promoting a systemic anti-inflammatory effect, in addition to increasing muscle protein synthesis and reducing peripheral muscle breakdown by reducing insulin resistance, as well as improving the energy status in muscle [33,34]. Exercise could be considered a weapon in the treatment of this condition.

### 5.2. Oxidative Stress

Another mechanism of interest in treating cachexia in cancer patients is redox homeostasis. In oxidative stress, there is an imbalance between the mechanisms that control oxidative production, causing an increase in reactive oxygen species (ROS) and a decrease in antioxidant mechanisms. This imbalance is present in chronic diseases [35] and may be associated with muscle loss through direct muscle action of ROS or through dysfunctions in the synthesis and degradation pathways of proteins, apoptosis, autophagy, and mitochondrial function [36].

Studies in animal models have already shown that exercise can balance redox homeostasis, combating the oxidative stress caused by cancer [37]. In addition, attenuation of muscle atrophy and restoration of its contractility [38], which contribute to the maintenance of muscle fiber quality, was also recorded. In humans, one meta-analysis of 19 studies on the antioxidant effects of exercise concluded that physical training, in healthy individuals or in patients with various diseases, was associated with a reduction in pro-oxidant parameters and an increase in antioxidant capacity [39].

### 5.3. Protein Homeostasis

Protein homeostasis is altered during cachexia, with an increase in muscle degradation and a reduction in muscle synthesis. Some studies suggest that exercise can reverse and/or attenuate proteolysis, as well as stimulate protein formation in patients with cancer-induced cachexia.

The mTOR pathway has an important part in cell proliferation and growth. Its activation is dependent on the IGF-1/PI3K/AKT pathway during exercise. The mTOR protein forms two complexes: mTOR Complex-1 (mTORC1), responsible for protein synthesis, and mTOR Complex-2 (mTORC2), involved in processes of cytoskeleton organization, cell survival, and metabolism [40,41]. In cancer-induced cachexia, resistance exercise stimulates the activation of mTORC1 and attenuates AMPK activity, which may induce protein synthesis [42], and mechanisms that activate the AKT/mTORC1 pathway can reverse the loss of strength and muscle mass in animal models [43].

On the other hand, a target for exercise reducing proteolysis in muscle is the UPP, described as responsible for muscle loss in cancer-induced cachexia [44]. In mouse models with heart failure, exercise counteracted ubiquitin–proteasome pathway overactivation and reduced skeletal myopathy [45]. In another study, in Wistar rats, low-intensity exercise prevented muscle atrophy induced by cancer cachexia through suppression of UPP and activation of the mTOR pathway [46].

Responsible for protein degradation in health muscles, the ALP is another pathway unregulated in cancer patients, leading to increased protein degradation [47] apparently without adequate protein clearance and to the accumulation of autophagosomes in cells [48]. In mice with colon carcinoma, aerobic exercise was able to restore basal levels of autophagy and improved muscle homeostasis [37,49], and combined exercises (aerobic and resistance training) improved muscle mass through regulation of autophagy and mitochondrial function [50].

### 5.4. Gene Regulation

Physical activity can induce several alterations in genes activation. Various epigenetic modifications can contribute to changes in gene expression in response to exercise, such as DNA methylation, post-translational modifications of histone proteins, and regulation of gene expression via specific miRNAs [51].

Peroxisome proliferator-activated receptor-γ coactivator (PGC)-1α, an important protein increased during exercise, activates several transcription factors and regulates the expression of peroxisome proliferator activated receptor-δ, nuclear respiratory factors, glucose transporter GLUT4, mitochondrial transcription factor A, and myocyte enhancer factor 2 [52,53]. During exercise, PGC-1α is demethylated and its levels are increased, allowing its involvement in the regulation of mitochondrial biogenesis, fatty and glucose metabolism, angiogenesis, antioxidant defense, and inflammation [54]. PGC-1α can suppress FoxO activity in skeletal and cardiac muscle, reducing muscle atrophy [55,56]. Figure 2 summarizes the changes that occur in muscle during exercise.

## 6. Exercise Modalities and Therapeutic Benefit in Cancer Cachexia

Most trials evaluate the benefits of two main modalities of physical exercise: endurance and resistance training. Endurance training (ET) is also known as aerobic or cardio exercise. It consists of activities where large muscles move in a rhythmic manner and for a sustained period of time, such as walking, running, and swimming [57]. It results in increases in heart rate and energy expenditure [58]. Resistance training (RT) promotes progressive overload to skeletal muscles, improving their strength and promoting hypertrophy [59]. Intensity, frequency, and repetitions may vary [57].

There are differences in the way both types of exercises activate metabolic pathways. In a Finnish trial, a marker gene expression was examined in muscle biopsies of 19 healthy men. ET induced responses typical of angiogenesis and mitochondrial biogenesis, and RT induced responses characteristic of angiogenesis and muscle hypertrophy [60]. RT can increase protein synthesis and activate the mTORC1 pathway, which seems to be an important mechanism for hypertrophy [53,61]. On the other hand, ET increases the activity of AMPK and calcium- and calmodulin-activated protein kinase, which, in turn, increase PGC1-α levels and mitochondrial biogenesis [52,53].

The practice of physical activity prior to tumor diagnosis and the development of cancer cachexia seems to prevent muscle loss resulting from this disease. In animal models with breast cancer, aerobic physical exercise, started before tumor injection and continued after it, demonstrated the ability to reduce tumor volume, in addition to preventing muscle dysfunction and atrophy caused by tumor cachexia [62,63]. In rats submitted to resistance exercises prior to tumor injection, cachexia was prevented through the attenuation of inflammation and oxidative stress caused by the neoplasm [64].

Although loss of muscle mass and strength are hallmarks of cancer-induced cachexia, RT alone does not appear to be sufficient to reduce this effect, challenging the logic of its use to develop muscle hypertrophy. In a study with murine models with tumor-induced cachexia, the isolated use of ET, despite not preventing weight loss, helped to maintain body function and rescued part of the muscle mass, while the isolated use of RT led to a faster physical deterioration [65]. In another study with Wistar rats, RT failed to reduce loss of muscle function, anorexia, or mortality rate [66]. A combination of ET and RT, however, seemed to be beneficial, with improved function and muscle mass in mice with C26 colon carcinoma [50].

Cardiac remodeling in response to exercise occurs due to enhanced PI3K/AKT/mTOR pathway, gene expression modulation, mitochondrial and metabolic adaptation [67]. In rats treated with doxorubicin, ET increased PGC-1α levels and prevented FOXO1 and MuRF-1 increase in cardiac muscle and FoxO3, MuRF-1, and BNIP3 increase in skeletal muscle, contributing to mechanisms of reduction of myopathy [55]. Regarding cardiac muscle loss induced by cachexia, aerobic exercise reduces cardiac remodeling and dysfunction in animal models. Mechanisms involved in this outcome include reduced necrosis and inflammation [68], reduced cardiac autophagy [69], and prevention of the increase of TNF-like weak inducer of apoptosis (TWEAK) levels and NF-κB activation [70]. The transcription factor NF-κB induces the expression of pro-inflammatory genes [71], increases proteolysis, and decreases the activity of PGC-1α, contributing to the cardiac remodeling present in cachexia [6], and ET can possibly prevent its activation by inhibiting TWEAK. More studies about cardiac modifications induced by exercise in cancer cachexia are necessary to understand all mechanisms it can affect, and evaluation of cachectic patients in trials including exercise may clarify the role of physical training to recover cardiac function.

Physical exercise has many positive effects in patients with cancer. A metanalysis of eight randomized controlled trials (RCTs) with exercise interventions (ET or RT or both) showed a reduced risk of mortality and recurrence in non-metastatic patients with cancer or in cancer survivors [72]. In non-cachectic patients with advanced or metastatic cancer, physical activity is feasible and maintains physical capacity, as demonstrated in RCTs with an association of ET and RT, even in older patients (≥65 years old) [73,74].

The role of exercise in patients with cancer cachexia is not well established. As previously described, the practice of physical activity seems to maintain body function and even increase muscle mass in cachectic animal models. One RCT evaluated the benefit of RT in 20 cachectic patients with head and neck cancer during radiotherapy and concluded that this is a feasible approach with possible improvement in fatigue and quality of life [75]. A recently published systematic review, composed of four RCTs including adult patients with cancer-induced cachexia, reported that the available studies so far are not able to answer whether physical exercise is effective, acceptable, or safe for this population, concluding that the existing evidence is of low quality [76].

## 7. Multimodal Approach in Cachexia

The isolated use of physical exercise does not seem to be enough to stop or reverse the muscle loss that occurs in cancer-induced cachexia. Thus, the studies currently carried out in this area aim at the combination of physical training with medication and/or nutritional supplementation, in order to have better results.

The rationale for using a multimodal approach to cachexia comes from the growing knowledge of the multiple metabolic pathways that contribute to this condition. As already described, increased inflammation and oxidative stress, reduced protein synthesis, and stimulation of muscle degradation pathways are part of the spectrum of cancer-induced cachexia, and targeting them through different therapies has the potential to be more effective.

The combination of physical exercise with drugs and/or nutritional supplements has been studied in cachectic animal models. A study in breast cancer mice evaluated the combination of selenium nanoparticles (Se NPs) supplementation with ET and the two approaches alone, demonstrating that the use of Se NP accelerated the development of cachexia, despite Se being an antioxidant, while the combination prevented muscle loss and showed the greatest reduction in tumor size among the groups evaluated [63]. In another study of eicosapentaenoic acid (EPA), the combination of this anti-inflammatory substance with ET led to a partial recovery of strength and muscle mass [77], although the use of EPA alone did not demonstrate any muscle effect in mice, which is in line with the lack of benefit in adult patients with cancer-induced cachexia seen in a previous RCT [78]. Finally, another investigation showed that the association of erythropoietin with ET helped to correct anemia resulting from cachexia and may be useful in preventing cardiac dysfunction resulting from this condition [79].

A multimodal approach seems feasible in patients with advanced cancer, as addressed by some studies. Combinations of physical exercise (ET and/or RT) with nutritional supplements containing omega-3 fatty acids (EPA or docosahexaenoic acid) [80,81] or branched-chain amino acids [82,83] or just nutritional counselling and dietary adjustments as needed [84] were safe approaches in this population, with some positive results in improving protein intake and muscle strength and reducing symptoms like nausea e vomiting. These results must yet be confirmed in more robust and phase III trials.

In cachectic cancer patients, there is still uncertainty about the benefit of combining physical exercise and other therapeutics in order to stop or reverse muscle loss. A trimodal approach is usually the preferred treatment studied in this population, which includes a combination of exercise, nutritional supplementation, and nonsteroidal anti-inflammatory drugs. The study ACCeRT was a small prospective trial in which 20 adult patients with non-small-cell lung cancer (NSCLC) and refractory cachexia were randomized to receive EPA and celecoxib or EPA, celecoxib, essential amino acids, and RT [85,86]. Interventions in both arms were feasible, and stabilization of body weight and free fat mass could be obtained for some weeks with both approaches [85]. In Pre-MENAC, a phase II trial with NSCLC and pancreatic cancer patients, celecoxib associated with EPA-enriched supplements and physical exercise (RT and ET) was a safe and viable intervention for pre-cachectic and cachectic individuals [87]. Although in both trials a trimodal approach was feasible, there is a lack of evidence on whether this therapy has any role in reducing or reverting skeletal or cardiac muscle degradation induced by cancer cachexia; more studies are needed to draw any conclusion.

Ongoing phase II e III randomized clinical trials in patients with advanced cancer, pre-cachexia, and/or cachexia may provide more information of the role of exercise/ multimodal therapies for treating this condition. These trials characteristics are described in Table 1.

## 8. Conclusions

Cancer-induced cachexia can be a complication of cancer progression which negatively affects the quality of life and overall survival of patients. Numerous metabolic pathways are modified in this condition, with increased inflammatory response and oxidative stress, decreased synthesis of proteins, and deregulation of genetic factors, resulting in weight and muscle mass loss, besides damage to several organs. Experiments in animal models suggest that physical exercises could counteract the affected pathways in cachexia. Training increases the levels of the anti-inflammatory myokine IL-6, regulates redox homeostasis, corrects mitochondrial biogenesis mediated by PGC-1α, stimulates protein synthesis, and reduces insulin resistance. All this could reduce skeletal muscle loss and cardiac muscle atrophy developed during cachexia.

Despite the benefits of exercise, its role in patients with cancer-induced cachexia is still uncertain. The best evidence of exercise feasibility and efficacy for muscle and/or weight recovery is found in patients in the pre-cachexia phase. In patients with consolidated cachexia, there are still few studies, and the results regarding clinical benefits do not allow us to state that this is a totally safe and useful approach for them. Furthermore, a multimodal treatment, with the use of nutritional supplements and anti-inflammatory drugs, associated with physical activity, seems currently to be the most accepted approach in the management of cancer-induced cachexia, but still lacks robust clinical studies to confirm its effectiveness in reducing muscle loss and weight and improving overall functionality in these patients.

## Figures and Tables

**Figure 1 cancers-14-00342-f001:**
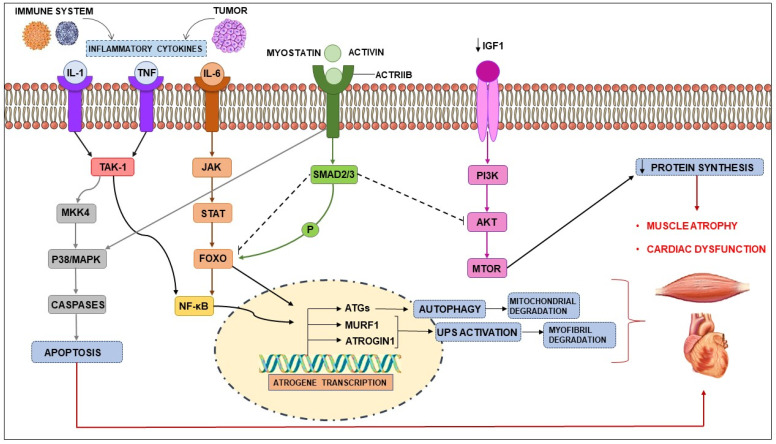
Signaling pathways involved in muscle wasting. Catabolic pathways of protein degradation: activation of inflammatory cytokines, particularly, TNF, IL-1, IL-6 leads to the activation of NF-κB and FOXO. The binding of IL-6 to its receptor induces STAT3 expression, which leads to the activation of the NF-κB pathway. The autophagy–lysosome system is activated by the transcription factor FOXO Activation of p38 and JAK/MAPK leads to apoptosis mediated by caspases. Myostatin can also activate protein degradation through FOXOs and may decrease protein synthesis, inhibiting AKT through SMAD. The levels of insulin-like growth factor-1 (IGF-1) are decreased during muscle wasting, suppressing the IGF-1 pathway and therefore inhibiting protein synthesis. The ubiquitin-proteasome system(UPS) is initiated by the transcription of the E3 ubiquitin ligases MuRF-1 and MAFbx/atrogin-1. ActRIIB, activin receptor type IIB; TNFα, tumor necrosis factor-α; IL, interleukin; JAK, Janus kinase; STAT, signal transducers and activators of transcription; FOXO, Forkhead box transcription factors; mTOR, mammalian target of rapamycin; NF-κB, nuclear factor-κB; MAPK, mitogen-activated protein kinase; MuRF-1, muscle RING-finger protein-1; IGF-1, Insulin-like growth factor-1; MAFbx, muscle atrophy F-box protein 1.

**Figure 2 cancers-14-00342-f002:**
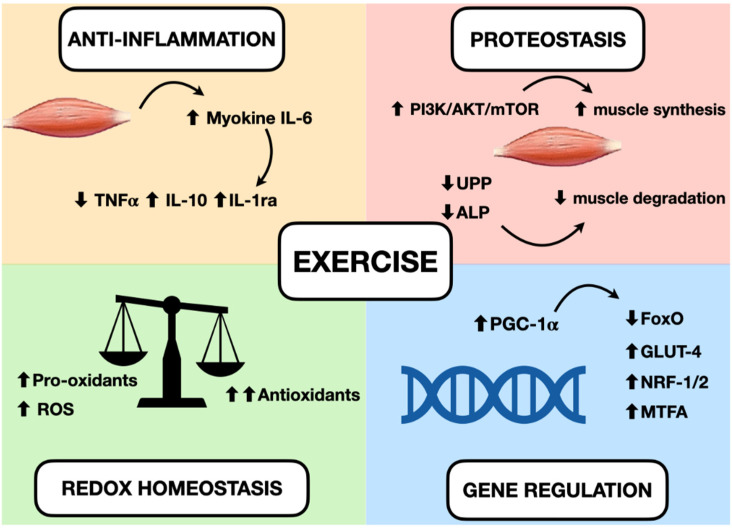
Effects of exercise in muscle and possible mechanisms for treating cachexia. Increase of myokine IL-6 can induce an anti-inflammatory effect, increasing IL-10 and IL-1ra and reducing TNF-alpha. A greater production of antioxidants than pro-oxidants can be responsible for restoring redox homeostasis during exercise. Increased activity of the PI3K/ALT/mTOR pathway and reduced activity of the ubiquitin–proteasome and autophagy–lysosomal pathways can induce protein homeostasis, increasing muscle synthesis and decreasing muscle degradation. The activation of PGC-1alpha can regulate genes involved in mitochondrial biogenesis and redox homeostasis (nuclear respiratory factors 1 and 2 and mitochondrial transcription factor A), increase the expression of GLUT-4, regulate glucose metabolism, and reduce FoxO function and proteolysis.

**Table 1 cancers-14-00342-t001:** Ongoing phase II and III randomized clinical trials of exercise intervention for cancer-induced cachexia.

Trial	Study Design	Characteristics of Included Patients	Physical Exercise Intervention	Other Interventions	Main Results
MENAC [88] Study Director: Kaasa S. and Fallon M.	RCT, phase III, open-label, multicenter	Inclusion criteria: NSCLC stage III or IV or pancreatic cancer stage III or IV due to commence first- or second-line anticancer therapy, KPS > 70	Interventional arm: functional resistance training three times each week and aerobic training twice a week.	Interventional arm: ONS with EPA and DHA, nutritional counselling, and ibuprofen Control arm: usual care	Identifier: NCT02330926 Primary outcome: change in body weight
MIRACLE Principal Investigator: Lee K.Y.	RCT, phase II, open-label	Inclusion criteria: GI and LC; first- or second-line chemotherapy; patients classified as normal, pre-cachexia, or cachexia	Interventional arm: Weekly physical exercise by physiatrist (60 min per visit)	Interventional arm: Ibuprofen, omega-3-fatty-acid, ONS, Bojungikki-tang, nutritional counselling, psychiatric intervention Control arm: conventional palliative care	Identifier: NCT04907864 Primary outcome: change in body mass and handgrip strength
NEXTAC-III Principal investigator: Naito T.	RCT, phase II, open-label	Inclusion criteria: ≥70 years old, local advanced or metastatic NSCLC or pancreatic cancer, cancer cachexia with an indication of anamorelin hydrochloride, new systemic chemotherapy (first-line in NSCLC and second-line in pancreatic cancer)	Interventional arm: home-based resistance training for 12 weeks	Interventional arm: Anamorelin hydrochloride, nutritional counselling	Identifier: JPRN-jRCTs041210053 Primary outcome: proportion of patients with a clinically meaningful reduction in 6 min walking distance

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
