# Peer review of "Impact of Cancer Cachexia on Cardiac and Skeletal Muscle: Role of Exercise Training"

_cancers, 2022, doi:10.3390/cancers14020342_

Round 1

Reviewer 1 Report

The manuscript of Bordignon et al. tried to introduce the influence of cancer cachexia on cardiac and skeletal muscle and whether physical exercise has any role in trying to prevent or reverse cancer cachexia. Though some recently published reports were included in the review, the review did not provide clear and well-classified information for readers.

Specific comments:

  1. Since the authors specifically mentioned cardiac muscle in the title, it is necessary to introduce the influence of cancer cachexia on cardiac muscle in detail and the possible molecular mechanisms. There are lots of studies and reviews about skeletal muscle wasting in cancer cachexia but fewer studies about cardiac muscle dysfunction in cancer cachexia. Provide more information about the similarity and difference between cardiac muscle and skeletal muscle wasting in cancer cachexia, especially the mechanisms.
  2. The Figure 1 is just the illustration about the mechanisms of muscle wasting in cancer cachexia, which could be found in other reviews introducing cancer cachexia. At least the signal pathways which might be affected by exercise should be shown in the illustration to support the aim of the review.
  3. Could the authors clearly summarize the possible mechanisms of exercise on muscle wasting in cancer cachexia? For example, there were reviews introducing the effects of exercise on autophagy (The Role of Autophagy Modulated by Exercise in Cancer Cachexia. Life (Basel) 2021 Aug 02;11(8)), inflammation (Exercise Training as Therapeutic Approach in Cancer Cachexia: A Review of Potential Anti-inflammatory Effect on Muscle Wasting. Front Physiol 2020;11; Exercise as an anti-inflammatory therapy for cancer cachexia: a focus on interleukin-6 regulation. Am J Physiol Regul Integr Comp Physiol 2020 02 01;318(2) ), metabolism (Exercise-A Panacea of Metabolic Dysregulation in Cancer: Physiological and Molecular Insights. Int J Mol Sci 2021 Mar 27;22(7)), and others. Please provide clear and well-classified summary of previous reports in a new review.

Reviewer 2 Report

This review paper aims to provide a general overview of how physical exercise training could be used to curb cachexia during cancer progression, with particular focus on cardiac and skeletal muscle. The papers discussed several mechanisms leading to muscle loss, including the inflammatory state induced by tumor-derived cytokines, the alterations in energy balance, and the signaling mechanisms leading to muscle protein breakdown. The paper also attempted to discuss the mechanisms leading to cardiac dysfunction. Finally, the paper provides key information as it relates to the beneficial effects of physical exercises on cancer cachexia. As such, this paper is very well written and provides a concise summary of the current advances in the cancer cachexia field. However, a few weaknesses were noted:

1) The paper will benefit from additional schematic figures describing the link between heart and muscle during cachexia as well as the pathways/mechanisms leading to cardiac dysfunction.

2) The therapeutic effects of exercise training on heart were barely discussed. This is a major aspect of this paper, and should be discussed accordingly.

3) It would be highly informative if a summary about the involvement of brain during exercise training is provided.

Reviewer 3 Report

Cachexia is one of the significant complications of cancer, which significantly impacts the survival outcome and quality of life in patients. Exercise training, which has established health benefits via metabolic and immune modulation, may potentially contribute to maintaining muscle function in patients. In this manuscript, the authors reviewed the mechanism of muscle atrophy and evaluated the effect of exercise in patients with cachexia.

Overall, the manuscript focused on a significant clinical challenge, which should significantly interest the readerships in the greater field. However, the manuscript must be improved to address the following major concerns:

The authors have summarized a network of signaling pathways involved in muscle wasting. However, much of the signaling pathways remain inaccurate or incomplete. For example, the transcriptional activity of FOXO is inhibited by phosphorylation. It is also speculative to suggest STAT3 signaling directly affects FOXO. On the other hand, the function of AKT as a known negative regulator of FOXO was not discussed. The quality of the discussion in terms of accuracy and completeness hindered the readability of the manuscript.

The authors have surveyed a handful of clinical trials assessing exercise as an intervention for cancer-induced cachexia. However, the quality of evidence and the implications of the studies remained questionable. It is also unclear whether the studies were assessed systematically, as per the Preferred Reporting Items for Systematic Reviews and Meta-Analysis (PRISMA) guidelines. Therefore, it remains uninformative as to the roles of exercise training on cancer cachexia, despite the effort to summarize the studies.

Round 2

Reviewer 1 Report

The manuscript has been improved. It is acceptable now.